# Cost-effectiveness of dengue vaccination in Puerto Rico

**Guido España [1]\*, Andrew J. Leidner [2], Stephen H. Waterman[3], T. Alex Perkins[1]**

**1** University of Notre Dame, Notre Dame, Indiana, United States of America, **2** Immunization Services Division, Centers for Disease Control and Prevention (CDC), Atlanta, Georgia, United States of America, **3** Dengue Branch, Centers for Disease Control and Prevention (CDC), San Juan, Puerto Rico, United States of America

\* guido.espana@nd.edu

**Data Availability Statement:** The code used in this study is available on the github repository: https://github.com/confunguido/puerto_rico_dengvaxia_cea. Data sources are publicly available and referenced in the methods section.

## Abstract

An effective and widely used vaccine could reduce the burden of dengue virus (DENV) around the world. DENV is endemic in Puerto Rico, where the dengue vaccine CYD-TDV is currently under consideration as a control measure. CYD-TDV has demonstrated efficacy in clinical trials in vaccinees who had prior dengue virus infection. However, in vaccinees who had no prior dengue virus infection, the vaccine had a modestly elevated risk of hospitalization and severe disease. The WHO therefore recommended a strategy of pre-vaccination screening and vaccination of seropositive persons. To estimate the cost-effectiveness and benefits of this intervention (i.e., screening and vaccination of seropositive persons) in Puerto Rico, we simulated 10 years of the intervention in 9-year-olds using an agent-based model. Across the entire population, we found that 5.5% (4.6%-6.3%) of dengue hospitalizations could be averted. However, we also found that 0.057 (0.045–0.073) additional hospitalizations could occur for every 1,000 people in Puerto Rico due to DENV-naïve children who were vaccinated following a false-positive test results for prior exposure. The ratio of the averted hospitalizations among all vaccinees to additional hospitalizations among DENV-naïve vaccinees was estimated to be 19 (13–24). At a base case cost of vaccination of 382 USD, we found an incremental cost-effectiveness ratio of 122,000 USD per QALY gained. Our estimates can provide information for considerations to introduce the CYD-TDV vaccine in Puerto Rico.

## Author summary

The first dengue vaccine, known as CYD-TDV, is currently under consideration as a public health tool in Puerto Rico, where dengue is endemic. Although the vaccine protects individuals with prior exposure to dengue virus, individuals vaccinated without prior dengue virus infection could experience a higher chance of severe disease. To avoid vaccinating individuals who have not been exposed to dengue virus, screening for prior infection before vaccination is recommended. Understanding the potential impacts of this intervention in specific settings can help public health decision makers to balance economic and health benefits to consider the implementation of this vaccine. In this study, we estimated

**Funding:** This publication was made possible with support from the NIH National Center for Advancing Translational Sciences (NCATS), awarded to the Indiana University School of Medicine, Postdoctoral Fellowship Training Grant Number TL1 TR002531, Thomas Hurley, PI, Indiana University School of Medicine, and UL TR002529 Anantha Shekhar PI, Indiana University School of Medicine, which provided funding support for TL1 Postdoctoral Fellowship appointee, GE. The funders had no role in study design, data collection and analysis, decision to publish, or preparation of the manuscript.

**Competing interests:** The authors have declared that no competing interests exist.

the impacts of incorporating screening and vaccination in Puerto Rico for the next 10 years. We found that the vaccine could have a positive impact in reducing severe dengue by preventing 5.5% of hospitalizations, but it could also cause 1.6 hospitalizations for every 1,000 vaccinated who had a false-positive screening result. We also found that this intervention could be cost-effective in Puerto Rico at a cost of 382 USD.

## Introduction

Dengue is considered one of the most important mosquito-borne viral diseases affecting humans, with half the world's population living in areas at risk [1,2]. Dengue virus (DENV) has been estimated to infect between 100 to 390 million individuals per year. Between 51 to 96 million of these infections result in disease episodes [1,3]. This disease has been of great concern in Latin America, which had a record number of dengue cases in 2019, with 2.7 million cases and 1,200 deaths reported to health systems [4]. In the absence of an effective vaccine against dengue, vector control has been the primary intervention available to reduce virus spread, but the evidence for the effectiveness of vector control to sustainably reduce dengue incidence is limited [5].

Currently, one dengue vaccine (CYD-TDV) is available and other vaccine candidates are showing promising results [6,7]. Recent analyses of CYD-TDV vaccine trials showed an increased risk of severe dengue upon subsequent natural infection among vaccinees without previous exposure to dengue virus [8]. In light of this finding, the World Health Organization (WHO) recommends pre-vaccination screening to help ensure that only those with previous exposure (DENV-exposed) are vaccinated [9]. The United States Food and Drug Administration licensed CYD-TDV for children 9–16 years living in dengue endemic areas and with documented evidence of previous dengue virus infection [10]. The requirement of a pre-vaccination laboratory screening for prior dengue virus infection complicates cost-effectiveness and logistical considerations for a vaccine program. The quality of pre-vaccination screening tests is also a critical consideration because test specificity can affect the number of false positive test results and because vaccinating a child with a false positive screening test result can put the child at risk for more severe dengue illness if infected. It is important therefore to estimate the potential health and economic benefits and risks of using CYD-TDV when paired with a pre-vaccination screening strategy.

Several studies have suggested that pre-vaccination screening with CYD-TDV could be cost-effective in some settings [11–14], but high specificity of screening is required to minimize individual risk, as well as high sensitivity to maximize population benefits [15]. Pre-vaccination screening is also recommended in Latin America [16]. The health benefits and cost-effectiveness of pre-vaccination screening for the Philippines and Brazil have been investigated in a recent study [12]. That analysis showed likely reduced disease incidence with limited adverse events at acceptable costs from this intervention with pre-vaccination screening and subsequent vaccination with CYD-TDV in areas with moderate to high-transmission intensity. Dengue is also endemic to several tropical and sub-tropical jurisdictions associated with the United States, including Puerto Rico [17]. A program of pre-vaccination screening coupled with CYD-TDV given to seropositive individuals is currently being considered to reduce the burden of dengue in Puerto Rico [18]. In this study, we evaluate the epidemiological and economic impacts of pre-vaccination screening and subsequent vaccination with CYD-TDV in Puerto Rico.

## Methods

We estimated the epidemiological benefits and cost-effectiveness of the intervention over a 10-year time-frame. We based this study on a previous analysis of the impact of this intervention in various transmission settings [12,19], and we adjusted transmission parameters and costs to represent Puerto Rico.

### Agent-Based model

To evaluate the impact of a pre-vaccination screening intervention in Puerto Rico, we modified an agent-based model of dengue transmission with humans and mosquitoes represented as agents. This model has been described in previous publications [12,19,20]. While our model has been calibrated to demographic and geographic data from Iquitos, Peru [20], the model can represent DENV transmission in a generic setting [19] and has been modified in this study to simulate DENV transmission dynamics in Puerto Rico. Our model compares two strategies, an intervention strategy and the status quo. The intervention strategy is the routine pre-vaccination screening and subsequent vaccination of seropositive of 9-year-olds in Puerto Rico. Nine-year-olds is the lowest age approved for vaccination, and has been used as a default age of vaccination in other studies [12,13,19]. For each of the strategies, the model population was followed for 10 years, keeping track of dengue-related health events defined as dengue virus infections, hospitalizations, and deaths.

### Model parameterization for Puerto Rico ($PE_9$)

We modified the transmission parameters of the model to approximate DENV transmission dynamics of DENV in Puerto Rico. We adjusted the transmission intensity of DENV in our model to achieve the expected age-specific dengue antibody prevalence levels ($PE_9$) in Puerto Rico. We estimated the $PE_9$ for Puerto Rico based on two serological studies. Coudeville et al. [21] estimated 50% seroprevalence in 9-year-olds in areas where the CYD-TDV phase-3 trials were conducted. Argüello et al. found that 49.8% (95% CI = 43.6–56.0%) of participants between 10–18 years of age had a positive IgG anti-DENV antibodies [22]. We assumed a baseline of $PE_9$ = 50%. Because of the potential negative outcomes in lower transmission settings, which could reduce the benefits of this intervention, and given the uncertainty in the estimates of $PE_9$, we explored a lower value of $PE_9$ = 30%. In addition, to assess the sensitivity of our estimates to higher intensity of transmission, we simulated an alternative scenario with slightly higher transmission intensity of $PE_9$ = 60% (Supplementary text).

Individuals can be infected with DENV four times over the course of their life. Evidence suggests the second infection can cause a disproportionate level of disease severity, relative to the first infection or to the third or later infections. In the model, the effect of a vaccination on an individual is similar to a natural infection without symptoms (silent infection), which is consistent with assumptions in other models [13,14,19,23]. If the individual is DENV-naïve, then a vaccination serves as the individual's first (a silent) infection, and a subsequent infection may be more severe. By contrast, if the individual is DENV-exposed, then a vaccination serves as the individual's second or later (silent) infection, which does not result in additional risk for severe disease. These assumptions are supported by empirical evidence showing that, compared to those not vaccinated, seronegative individuals vaccinated with CYD-TDV have a higher risk of hospitalization after a natural infection, whereas seropositive individuals vaccinated have a lower risk of hospitalization [8]. Vaccine effectiveness parameters were calibrated to the most recent dengue vaccine trial results [8], we calibrated model parameters characterizing vaccine profile to vaccine trial data using a particle filtering approach, which is explained

**Table 1. Vaccine profile parameters calibrated to CYD-TDV data on vaccine efficacy [8].**

| Description | Fit | Lower 95% CI | Upper 95% CI |
|---|---|---|---|
| Average duration (days) of protection against infection for seronegative vaccinations | 426.69 | 41.02 | 733.75 |
| Average duration (days) of protection against infection for seropositive vaccinations | 258.66 | 136.71 | 464.09 |
| Probability of vaccine protection against infection for seronegative vaccinations conditional on exposure | 0.32 | 0.05 | 1.00 |
| Probability of vaccine protection against infection for seropositive vaccinations conditional on exposure | 0.52 | 0.15 | 0.97 |
| Probability of symptoms conditional on primary infection | 0.41 | 0.26 | 0.54 |
| Probability of symptoms conditional on secondary infection | 0.34 | 0.27 | 0.52 |
| Probability of symptoms conditional on post-secondary infection | 0.09 | 0.04 | 0.13 |
| Probability of hospitalization conditional on symptoms from primary infection | 0.07 | 0.04 | 0.11 |
| Probability of hospitalization conditional on symptoms from secondary infection | 0.38 | 0.27 | 0.42 |
| Probability of hospitalization conditional on symptoms from post-secondary infection | 0.10 | 0.06 | 0.11 |

in more detail elsewhere ([12], Appendix S2). The results from the calibration step are shown in Table 1 with the upper and lower bounds of the 95% confidence interval.

Given the stochastic nature of our model, we simulated 3,000 paired replicates over the parameter ranges, and reported the smoothed output of these simulations using a generalized additive model (GAM) in R. The uncertainty on the model parameters estimated in Table 1 was taken into account by simulating the model over the upper and lower bound of the estimates of the vaccine profile parameters.

## Epidemiological outcomes

Population-level benefits were defined as the proportion of symptomatic and hospitalized cases averted for the total population in comparison with a scenario without vaccination. Individual-level benefits were defined as the relative risk for symptomatic and hospitalized dengue of an individual child who undergoes screening and possibly vaccination compared to a child who is not given a screening or any vaccination. Since misclassification by a serologic test could result in DENV-naïve individuals receiving a vaccination that could lead to an episode of hospitalization [8], we quantify the magnitude of this risk by estimating the proportion of additional hospitalizations that occurred in the model among misclassified DENV-naïve individuals who received a vaccination. We reported the number of hospitalizations in vaccinated DENV-naïve children per 1,000 children vaccinated. We also reported the ratio of averted hospitalizations among the total population divided by the number of additional hospitalizations among misclassified DENV-naïve individuals who received a vaccination. This ratio captures the trade-off between hospitalizations that are averted by vaccination and any additional hospitalizations that may be caused by the vaccination of misclassified DENV-naïve individuals. Higher values of this ratio represent better value in terms of averted hospitalizations. Finally, we reported the number of additional hospitalizations for every 1,000 population.

In our simulations, we assumed that coverage of the intervention (i.e., serological screening and vaccination in the event of a positive result) was given to 80% of the target population, but

evaluated an alternative scenario of lower coverage (50%) in the sensitivity analysis. We assumed that 100% of children with a positive screening result were vaccinated. The baseline values of sensitivity (0.8) and specificity (0.95) were based on a recent review of rapid diagnostic tests for determination of serostatus [24]. Similar values were found by the dengue branch surveillance and research laboratory [25]. Assuming that increasing sensitivity would result in lower specificity, and vice-versa, we assumed three additional scenarios: 1) high sensitivity (0.95) and low specificity (0.76), 2) low sensitivity (0.64) and high specificity (0.95), 3) high sensitivity (0.95) and high specificity (0.95). In a set of sensitivity analyses, we also simulated pre-vaccination screening over the full range of values of sensitivity and specificity (0–1) to find the minimum values required to achieve positive proportion of dengue cases averted.

## Cost-effectiveness analysis

We evaluated the incremental cost-effectiveness ratio (ICER) of the intervention over a time horizon of 10 years. We used a public health perspective for our cost-effectiveness analysis, which included the costs of vaccination, screening, disease outcomes paid by government, as well as the quality-adjusted life-years and utility loss associated with premature death. Although this perspective could underestimate the cost-effectiveness from a societal perspective, we focused on a public health perspective to provide estimates for decisions makers in the public health sector and the health sector overall. This perspective has been used in previous economic analyses of the potential impact of routine vaccination with CYD-TDV [12,19]. We calculated the ICER as shown in Eq 1, which compares an intervention strategy to a no-intervention strategy. In the denominator, three different outcome measures are used to quantify the effectiveness of each strategy ($E_{intervention}$ and $E_{no-intervention}$): QALYs, symptomatic cases with medical care, and hospitalizations.

$$ICER = \frac{Cost_{intervention} - Cost_{no-intervention}}{E_{intervention} - E_{no-intervention}}.$$  (1)

To evaluate the ICER, we identified a set of cost parameters. The baseline cost per fully vaccinated child was set to 382 USD (32 USD—682 USD) based on the average current price of vaccines in the US per dose that had a similar age range, recombinant technology, and recent approval (S1–S4 Tables). The range of the vaccine cost was chosen based on the minimum and maximum values observed in the vaccines included (10 USD per dose and 227 USD per dose). The individual cost of screening was set to 30 USD, but we varied this price from 1 USD to 60 USD in sensitivity analyses.

Estimates of the costs paid by the government associated with treatment of dengue cases for ambulatory cases and hospitalizations were based on estimates from 2002 to 2009 (projected to 2010) in Puerto Rico [17]. Using the consumer price index for medical care for Puerto Rico [26], we adjusted these costs from 2010 values to 2019 USD. The cost of an ambulatory case was set to 315 USD (252–378) and the cost of a hospitalization was set to 2,132 USD (1,705–2,558). Future costs were discounted by 3% annually.

For the cost-effectiveness results, our primary health outcome was the quality-adjusted life-years (QALYs), which is a health-related quality of life index that ranges from 0.0 to 1.0 with 0.0 representing a state of death and 1.0 representing perfect health for one year. Following approaches used in previous studies of dengue [19,27–29], we estimated the QALYs gained from the intervention based on lost quality of life due to dengue fever and severe dengue. We used disutility parameters (D), or QALY decrements, that were taken from a study that estimated the quality of life associated with dengue fever and hospitalizations, conducted by Zeng et al. [30]. These values are listed in Table 2. Assuming a discount rate (r) of 3%, we estimated

**Table 2. Disutility for dengue cases.**

| Parameter | Value | 95% CI | Reference |
|---|---|---|---|
| Disutility (dengue fever) | 0.0307 | 0.0170–0.0917 | [30] |
| Disutility (hospitalizations) | 0.0351 | 0.0241–0.0960 | [30] |
| Disutility (death) | 1 | - | - |

the QALYs, such that

$$QALYs = \sum_{t}^{T}\left(\frac{\sum_{i}^{N}(1 - \sum_{j}^{J} D_{i,j,t})}{(1 + r^t)}\right), \tag{2}$$

where $i$ is each individual in the model, $j$ is the disease state (dengue fever, hospitalized dengue, or death), N is the total population in the model, and T is the number of years of the intervention (10 years).

To account for utility loss associated with premature death caused by severe dengue, we approximated remaining lifetime QALYs based on life expectancy (L). Life expectancy was obtained from the demographic characteristics of the modeled population and varied for each individual in the simulation. Discounted QALYs are estimated over the remaining lifetime of each individual in the population using a 3% discount rate. These remaining lifetime QALYs were calculated as

$$remaining\ lifetime\ QALYs = \sum_{i}^{N}\left(\frac{1}{r} - \frac{1}{r(1 + r)^{Q(L - \alpha_i)}}\right), \tag{3}$$

where the terms not already defined in the discussion of Eq 2 are defined as follows: Q is the future value of a QALY for all individuals each year who do not die from dengue, assumed to be 1.0 in all years beyond the simulation time horizon, L is the expected lifespan of an individual in the model, and $\alpha_i$ is the age of individual $i$ at the end of the model horizon.

## Results

### Epidemiological outcomes

**Population-level benefits.** From a population level, our simulations showed that the intervention resulted in averted symptomatic cases across all four scenarios for different levels of sensitivity and specificity of pre-vaccination screening tests (Fig 1, left panel). Estimates of the numbers of symptomatic cases and hospitalizations averted for each scenario of transmission intensity are shown in Table 3. More than 3,000 hospitalizations were averted in the baseline scenario and in the four scenarios that investigated different levels of prior exposure and pre-vaccination screening test properties (Fig 1, right panel) with 5.5% (4.6% - 6.3%) of hospitalizations averted when $PE_9 = 50\%$, and 3% (2% - 3.6%) when $PE_9 = 30\%$ (Fig 1, right panel). Higher sensitivity (0.95) and lower specificity (0.76) resulted in lower proportions of hospitalizations averted. Lower sensitivity (0.64) and higher specificity (0.99) also lowered the proportion of hospitalizations averted to 4.7% (3.8% - 5.3%) when $PE_9 = 50\%$, and to 2.8% (2.1% - 2.8%) when $PE_9 = 30\%$. Finally, increasing both, sensitivity (0.95) and specificity (0.95), increased benefits of vaccination up to 6.2% (6% - 7%) in the baseline transmission scenario ($PE_9 = 50\%$).

A reduction of symptomatic cases was achieved under any value of specificity and sensitivity for the pre-vaccination screening laboratory test (Fig 2). With regard to hospitalizations, our simulations over the whole range of sensitivity and specificity values suggest that even with

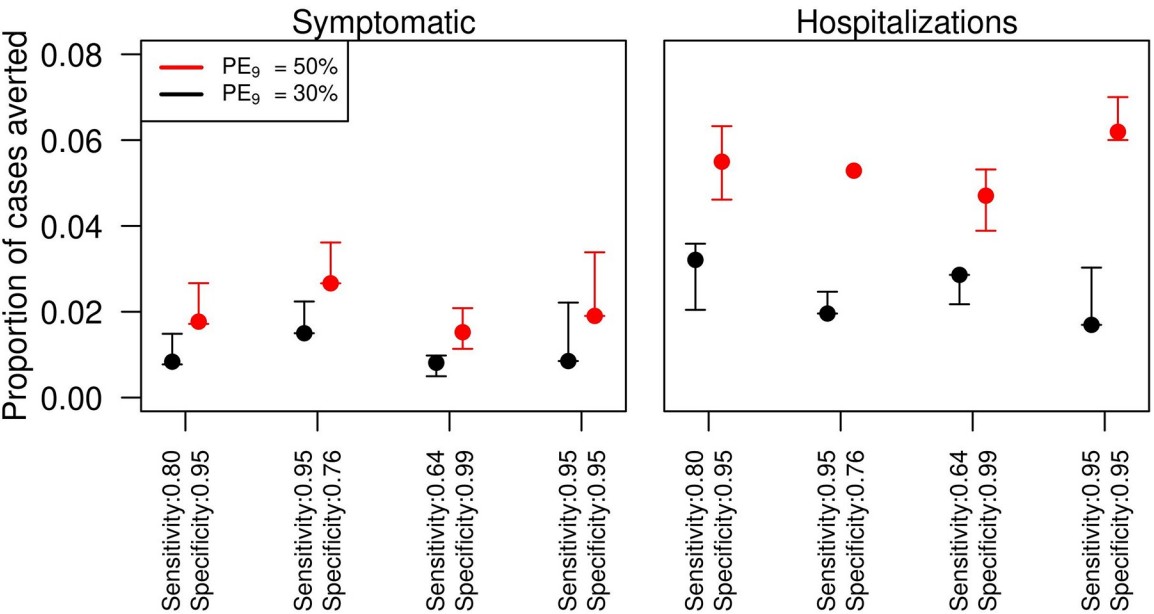

**Fig 1. Proportion of total dengue cases averted in Puerto Rico with the pre-vaccination screening strategy in 9-year-olds for different values of the sensitivity and specificity of pre-vaccination screening test.** Left panel refers to symptomatic cases and right panel to hospitalizations. The x-axis shows different assumptions on the specificity and sensitivity of the pre-vaccination screening test. The simulations were performed for 80% intervention coverage of routine pre-vaccination screening and subsequent vaccination in the event of positive result for 9 year-olds over 10 years.

perfect sensitivity, the minimum value of specificity to avert hospitalized cases is 0.6 for $PE_9$ of 30% (Fig 2, right panel). Holding the baseline level of specificity (0.95) constant, we found that sensitivity values above 0.2 result in positive cases averted. These specificity values (0.2 $PE_9$ = 50%, 0.6 $PE_9$ = 30%) represent the absolute minimum test parameters needed in the model to avoid an increase in hospitalizations.

## Individual-level benefits

The relative risk of becoming a symptomatic case was slightly reduced among vaccinated sero-positive individuals, compared to individuals of the same age who were not given the intervention (Fig 3, left panel). In the baseline scenario of sensitivity and specificity of laboratory test screening, the risk was around 0.85 (0.82–1.0) for $PE_9$ = 50% and around 0.9 (0.87–1.0) for $PE_9$ = 30%. The relative risk of a hospitalization was also reduced to around 0.63 (0.3–1.0) for $PE_9$ = 50% and to 0.73 (0.73–1.0) for $PE_9$ = 30% (Fig 3, right panel).

## Magnitude of naïve children at increased risk of hospitalization due to vaccination

We estimated the number of hospitalizations due to false positives of DENV-naïve children who were therefore vaccinated. We estimated these hospitalizations to be around 1.6 (1.3–2.1)

**Table 3. Estimates of symptomatic cases and hospitalizations averted in Puerto Rico.**

| Prior exposure 9yrs | Baseline symptomatics | Baseline hospitalizations | Averted symptomatics | Averted hospitalizations | Additional hospitalizations |
|---|---|---|---|---|---|
| 0.300 | 225,460 | 51,790 | 1,886 | 1,662 | 214 |
| 0.500 | 262,852 | 62,113 | 4,652 | 3,415 | 184 |
| 0.600 | 275,317 | 64,571 | 6,377 | 4,664 | 164 |

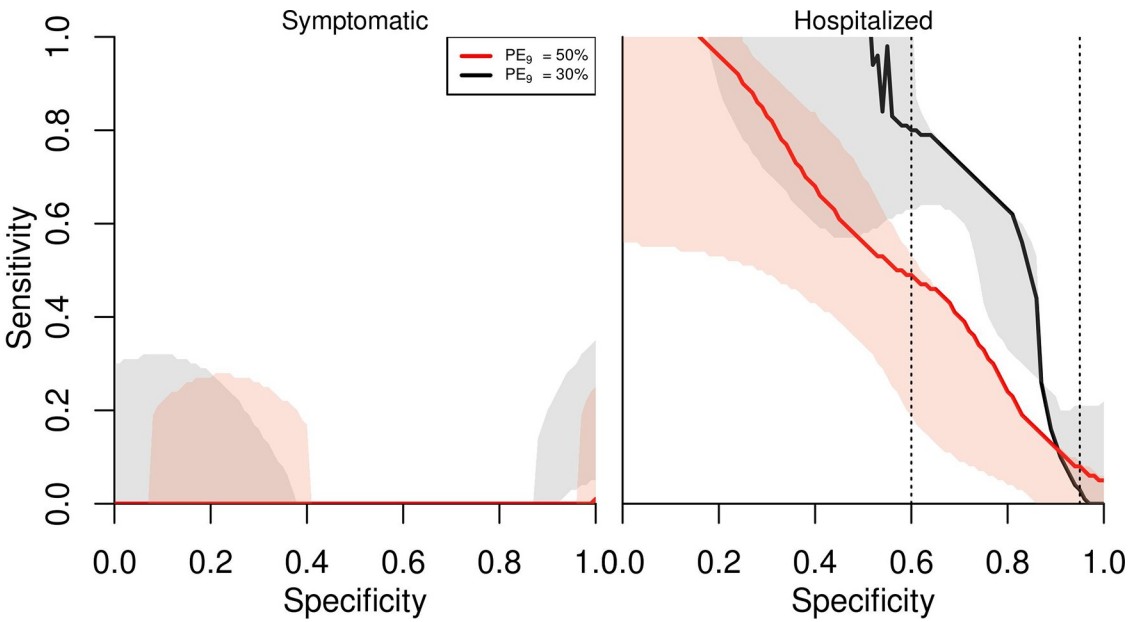

**Fig 2. Minimum sensitivity required for each specificity value to have positive averted cases with pre-vaccination screening strategy.** Left panel refers to symptomatic cases and right panel to hospitalizations. The shaded areas in each line represent the lower and upper bounds of the model parameters adjusted to reproduce the vaccine trial results. Values of sensitivity and specificity above curves show reduction of symptomatic or hospitalized cases. The simulations were performed for 80% intervention coverage of routine pre-vaccination screening and subsequent vaccination in the event of positive result for 9 year-olds over 10 years.

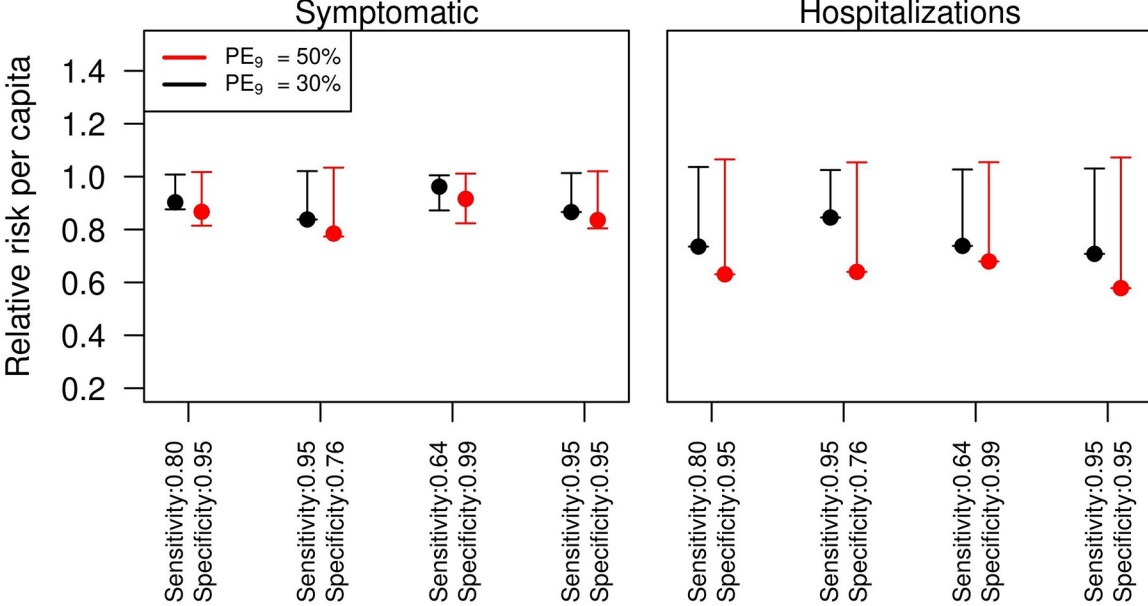

**Fig 3. Relative risk of dengue health outcomes for individuals in the intervention group.** Left panel refers to the risk of symptomatic cases, and the right panel to the risk of hospitalizations. Red lines show an assumed intensity of transmission of $PE_9 = 50\%$, and black lines represent $PE_9 = 30\%$. The simulations were performed for 80% intervention coverage of routine pre-vaccination screening and subsequent vaccination in the event of positive result for 9 year-olds over 10 year time horizon.

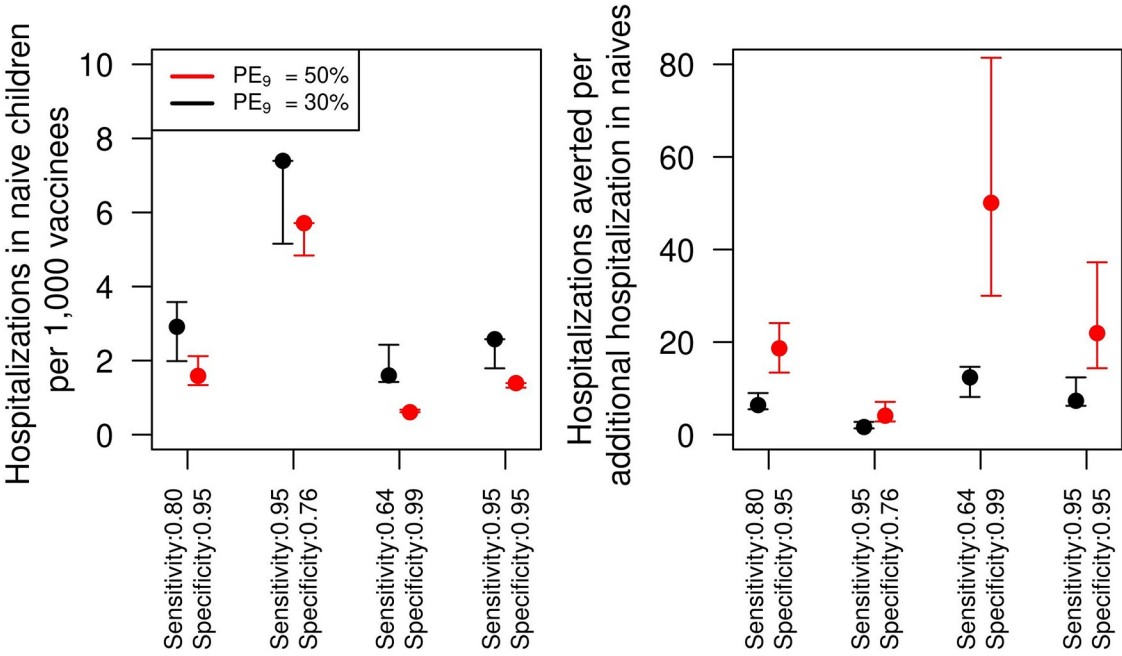

**Fig 4. Number of additional hospitalization cases due to vaccination of DENV-naïve children and ratio of hospitalizations averted to additional hospitalizations at different levels of sensitivity and specificity.** Left panel shows the number of hospitalizations per every 1,000 children vaccinated. The right panel shows the number of hospitalizations averted for each additional hospitalization in the DENV-naïve group. The simulations were performed for 80% intervention coverage of routine pre-vaccination screening and subsequent vaccination in the event of positive result for 9 year-olds over 10 year time horizon.

per 1,000 vaccinees in the baseline scenario of sensitivity (0.8) and specificity (0.95) ($PE_9$ = 50%, Fig 4, left panel). This number increased to almost 3 (1.9–3.6) cases per 1,000 vaccinees in the lower transmission scenario. Reducing specificity to increase sensitivity resulted in more than double the baseline hospitalizations in the DENV-naïve group. In contrast, reducing sensitivity to increase specificity to 0.99 reduced the number of hospitalizations to about half the baseline number. We also found that for every one additional hospitalization in the group of vaccinated DENV-naïve individuals there were about 19 (13–24) ($PE_9$ = 50%) hospitalizations averted overall in the baseline scenario (Fig 4, right panel). In the lower transmission setting, the ratio of averted hospitalizations among DENV-exposed individuals to additional hospitalizations among DENV-naïve individuals declined to around 7.7 (5.5–9.0). In proportion to the overall population, we estimated a total of additional hospitalizations of 0.057 (0.045–0.073) per 1,000 population in the baseline setting of transmission. This number increased to 0.067 (0.054–0.087) per 1,000 population at a lower transmission setting.

The number of hospitalizations due to misclassification was reduced further in an alternative scenario of higher transmission intensity (S1 Fig). Similarly, there was an increase in the number of hospitalizations averted in those with previous exposure to DENV (Supplementary text).

## Cost-effectiveness of the pre-vaccination screening intervention

We evaluated the incremental cost-effectiveness ratio (ICER) for the baseline scenario of costs, with a vaccine cost of 382 USD and screening of 30 USD. Under this scenario, the ICER of the intervention was around 122,000 USD per QALY gained (74,000–182,000) (Fig 5, left panel). The ICER was 240,000 USD at a lower transmission intensity scenario ($PE_9$ = 30%). At the

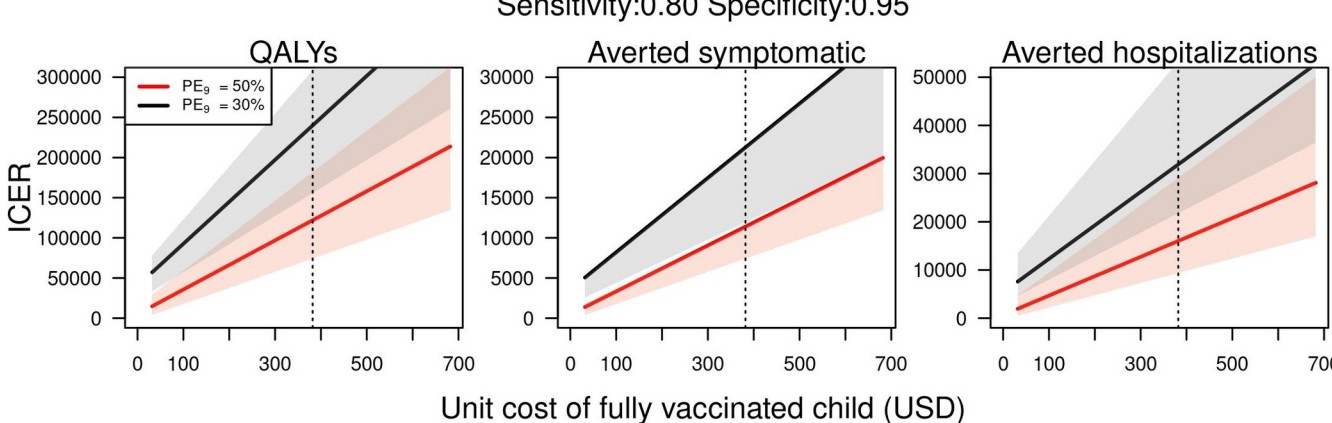

**Fig 5. ICER of pre-vaccination screening strategy in Puerto Rico at different costs of vaccination (total cost for three doses per person), assuming a unit cost of serological screening of 30 USD.** Dotted line represents the baseline assumption of vaccine cost (382 USD). All costs in 2019 USD.

minimum vaccine price of 32 USD, we found an ICER of 15,000 USD and 57,000 USD for the moderate and low transmission scenarios, respectively. In sensitivity analyses on vaccine cost, we find that for each additional dollar of vaccine cost, the ICER increased by 305 USD ($PE_9$ = 50%) and 520 USD ($PE_9$ = 30%), depending on transmission intensity. At a higher cost of serological screening of 60 USD, we found that the ICER increased to 143,000 USD ($PE_9$ = 50%) and 297,000 USD ($PE_9$ = 30%). In terms of the ICER per averted symptomatic case, we estimated that the intervention costs around 11,000 USD to avert a symptomatic case at moderate transmission (Fig 5, middle panel), and around 21,000 USD at a lower transmission setting. Finally, the cost to avert a hospitalized case was around 16,000 USD for a moderate transmission scenario and around 32,000 USD at a lower transmission scenario (Fig 5, right panel).

## Discussion

Using an agent-based model of dengue virus transmission, we simulated the impact of a routine program of pre-vaccination screening of 9-year-olds with subsequent vaccination of seropositive individuals, at dengue transmission levels that were calibrated to Puerto Rico. Assuming a moderate and low transmission intensity ($PE_9$ = [50%, 30%]) in Puerto Rico, we found that this intervention could be beneficial at the population-level and less so at the individual-level, as long as the serological pre-vaccination screening test has at least moderate values of sensitivity and high specificity. In sensitivity analyses, we found that this intervention could be more beneficial and cost-effective in the event that conditions of transmission in Puerto Rico are higher ($PE_9$ = 60%) than our baseline assumption. We also found that a minimum specificity of 0.6 for serological screening was required to ensure that the intervention would not result in an increase of hospitalizations due to vaccination of DENV-naïve individuals misclassified with a positive result for prior exposure. Additional epidemiological impact from different scenarios of transmission intensity can be explored in our webapp (http://denguevaccine.crc.nd.edu). In a cost-effectiveness analysis under our baseline assumptions, we estimated an ICER of around 122,000 USD per QALY gained. This value was sensitive to changes in the underlying transmission intensity and uncertainty in the accuracy and cost of serological screening, and uncertainty in vaccination costs.

A sensitivity analysis showed that higher specificity would be more important than high sensitivity for epidemiological benefits as well as cost-effectiveness. This, ensuring highly specific pre-vaccination screening tests to minimize the number of seronegative individuals

misclassified as seropositive is clearly important. Additional hospitalizations caused by vaccinating misclassified DENV-naïve children is a unique and important issue for the use of CYD-TDV. In our baseline scenario, our results suggest that for every 1,000 vaccinated children, around 2 extra cases would be caused (4 in a lower transmission setting). This number more than doubled when we adjusted the parameters of the pre-vaccination screening test, reducing the specificity of screening from 0.95 (with sensitivity 0.8) to 0.76 (with sensitivity 0.95).

We focused on a scenario where 80% of 9-year-olds were routinely given pre-vaccination screening and subsequent vaccination of seropositives. However, this level of intervention coverage (80%) may not occur in practice. There are many possible reasons that widespread use of pre-vaccination screening and any subsequent vaccinations may not be achieved. A number of challenges, common to many vaccines, may potentially reduce intervention coverage, such as: (1) the vaccine and screening tests may not be given a strong recommendation or endorsement from the medical community, (2) concerns of patients and parents over the perceived benefits and perceived safety of the vaccination, (3) costs of the pre-screening tests and vaccination, and (4) any number of logistical challenges that could occur with a complex intervention procedure that includes a pre-vaccination screening test that is potentially followed up with three doses of a vaccination, with each dose given approximately 6 months apart. Our results suggest that the cost-effectiveness could be affected by a lower uptake, slightly increasing the ICER, especially for QALYs gained and averted symptomatic cases in low transmission scenarios.

Generally, our results are consistent with other studies that evaluated dengue vaccination in Puerto Rico. Some of the differences in the results appear to be due to differences in important assumptions. At a vaccine cost of 20 USD per dose (60 USD for three doses), Coudeville et al. found that, using the GDP per capita as a threshold for the willingness to pay for a DALY (Disability-Adjusted Life Year) averted, this intervention could result in cost-savings [13]. One important difference is that our study used a higher cost of vaccination in the baseline scenario. At the lower bound of vaccination cost assumed in our study (32 USD for three doses), the cost per QALY gained was between 15,000 USD and 57,000 USD. Similar results were found by Zeng et al., although they simulated universal vaccination without screening [31]. Our study differs in the assumption of the costs of clinical care. We focused on the average cost paid by the government (1,615 USD) [17], the two studies mentioned above [13,31] used the overall direct cost of hospitalization (4,135 USD), which includes the cost paid by insurance, households, and employers. While these could account for some differences between the results of our studies, our sensitivity analyses suggest that changes to health care costs would only modestly affect the ICER estimates. At the base case of vaccine cost of 382 USD, dengue vaccination was estimated to cost 122,000 USD per QALY gained. Our estimate for the cost-effectiveness of dengue screening combined with vaccination falls within the range of economic values that have been estimated for some of the other adolescent vaccines in the United States [32]. For example, the cost per QALY gained for influenza vaccination of adolescents 12–17 years who are not at high risk was 119,000 in 2006 USD [32].

Although our model has been carefully parameterized our approach has some limitations. First, we did not explicitly simulate demographic characteristics of Puerto Rico. Instead, we parameterized our model to represent generic settings of transmission intensity with a demographic structure appropriate to Iquitos, Peru. The assumption of similar age-structures can result in some differences in transmission patterns. Nonetheless, in previous analyses with this model, our approach showed similar projections on the impact of dengue vaccination to seven other models [19]. In addition, we assumed that prevalence was similar across regions of Puerto Rico based on studies of seroprevalence [22]. Although comprehensive seroprevalence data by municipalities in Puerto Rico is not currently available, less populated municipalities

would likely have lower prevalence than densely populated urban areas. In consequence, our assumption of $PE_9 = 50\%$ would likely cover the majority of children who live in densely populated areas. Ideally, dengue seroprevalence studies could be performed to measure prevalence in less densely populated areas before introduction of the vaccine. Nonetheless, our results show that this intervention has epidemiological and economic benefits even in lower transmission scenarios ($PE_9 = 30\%$), as long as specificity of screening is high.

Another limitation of our model is that we are simulating homogeneous circulation of DENV serotypes. Although this assumption is unrealistic, given that dengue outbreaks are characterized by the dominance of one of the serotypes, it would be infeasible to make projections of the serotype-specific DENV importations for the next ten years. Under a scenario in which vaccine efficacy is balanced across serotypes, our estimates would likely remain robust to assumptions on the dominance of serotypes. However, in the event that transmission in Puerto Rico were to be dominated by a specific serotype, and vaccine efficacy were lower for that specific serotype, our estimates could overestimate the impact of CYD-TDV vaccination. In our baseline parameter for the cost of a vaccinated child, we did not explicitly include some factors such as transportation costs for the child and a caregiver, and storage and distribution of vaccine materials. These additional factors would likely serve to increase the cost per capita of a fully vaccinated child relative to our baseline assumption, which is the reason for the extensive sensitivity analyses included around vaccine costs and for the relative broad ranges applied to this parameter.

In summary, we estimated the cost-effectiveness of serologic screening and subsequent vaccination on the range of 74,000 to 182,000 USD per QALY, with a range dependent on assumptions about background prevalence, vaccine cost, and other factors. Model results comport reasonably well with other epidemiological models and with other estimates of economic value. Our analysis suggests that epidemiological benefits can be achieved from this intervention in Puerto Rico for the baseline assumptions of sensitivity and specificity. If multiple diagnostic tests were to be available, the specificity of the test should be prioritized given that it is crucial to minimize the additional hospitalizations in vaccinees without previous exposure to DENV. In conclusion, decision makers in Puerto Rico considering the implementation of this vaccine need to contemplate the potential benefits and risks of the intervention at the individual and population level. Our detailed computational model can provide relevant information that can be used to support decision making.

## Supporting information

**S1 Text. Sensitivity Analysis.**
(DOCX)

**S1 Fig. Number of additional hospitalization cases due to vaccination of DENV-naïve children and ratio of hospitalizations averted to additional hospitalizations at different levels of sensitivity and specificity.** Left panel shows the number of hospitalizations per every 1,000 children vaccinated. The right panel shows the number of hospitalizations averted for every additional hospitalization case in the DENV-naïve group. The simulations were performed for 80% intervention coverage of routine pre-vaccination screening in 9 year-olds over 10 years. (TIF)

**S2 Fig.** ICER of pre-vaccination screening strategy in Puerto Rico with a higher transmission setting (PE9 = 60%) at different costs of vaccination (total cost for three doses per person), assuming a unit cost of serological screening of 30 USD. Red lines represent a transmission intensity scenario of PE9 = 50%, and black lines represent a transmission scenario of

PE9 = 30%. All costs in 2019 USD.
(TIF)

**S3 Fig. ICER of pre-vaccination screening strategy in Puerto Rico at different cost of vaccination (3 doses per person), assuming a unit cost of serological screening of 30 USD.** Dotted vertical line represents the baseline cost of vaccination (382 USD). All costs in 2019 USD.
(TIF)

**S4 Fig. ICER of pre-vaccination screening strategy in Puerto Rico at lower coverage (50%) and at different cost of vaccination (3 doses per person), assuming a unit cost of serological screening of 30 USD.** Solid lines shows the baseline scenario of coverage (80%) and dashed line shows lower coverage assumption (50%). All costs in 2019 USD.
(TIF)

**S1 Table. Sensitivity to changes in costs and disutility for ICERs measuring costs (2019 USD) per QALYs gained.**
(DOCX)

**S2 Table. Estimated price for CYD-TDV based on age range, recombinant technology, and recent approval.**
(DOCX)

**S3 Table. List of vaccines recommended for 7 to 15 year-olds.**
(DOCX)

**S4 Table. List of vaccines with similar technology as CYD-TDV.**
(DOCX)

**S5 Table. List of vaccines introduced in the last 5 years.**
(DOCX)

## Acknowledgments

The findings and conclusions in this report are those of the authors and do not necessarily represent the views of the Centers for Disease Control and Prevention.

## Author Contributions

**Conceptualization:** Guido España, Andrew J. Leidner, Stephen H. Waterman, T. Alex Perkins.

**Data curation:** Guido España.

**Formal analysis:** Guido España, Andrew J. Leidner, T. Alex Perkins.

**Methodology:** Guido España, Andrew J. Leidner, T. Alex Perkins.

**Supervision:** Andrew J. Leidner, Stephen H. Waterman, T. Alex Perkins.

**Validation:** Guido España.

**Visualization:** Guido España.

**Writing – original draft:** Guido España, Andrew J. Leidner, Stephen H. Waterman, T. Alex Perkins.

**Writing – review & editing:** Guido España, Andrew J. Leidner, Stephen H. Waterman, T. Alex Perkins.

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
