## [Decision Letter · Decision Letter 0]

25 Feb 2021

Dear Mr. España,

Thank you very much for submitting your manuscript "Cost-effectiveness of Dengue Vaccination in Puerto Rico" for consideration at PLOS Neglected Tropical Diseases. As with all papers reviewed by the journal, your manuscript was reviewed by members of the editorial board and by several independent reviewers. The reviewers appreciated the attention to an important topic. Based on the reviews, we are likely to accept this manuscript for publication, providing that you modify the manuscript according to the review recommendations. 

Sincerely,

Donald S Shepard

Associate Editor

Sergio Recuenco

Deputy Editor

Reviewer's Responses to Questions

**Key Review Criteria Required for Acceptance?**

**Methods**

-Are the objectives of the study clearly articulated with a clear testable hypothesis stated?

-Is the study design appropriate to address the stated objectives?

-Is the population clearly described and appropriate for the hypothesis being tested?

-Is the sample size sufficient to ensure adequate power to address the hypothesis being tested?

-Were correct statistical analysis used to support conclusions?

-Are there concerns about ethical or regulatory requirements being met?

Reviewer #1: This is an economic analysis of the currently favored strategy for Dengue vaccination, namely vaccination at 9 years of age or above of children who are seropositive for Dengue, but not children who are seronegative (the former are protected, but the latter are put at enhanced risk of serious illness by vaccination). The model is applied to Puerto Rico, although data from Peru were used to develop it.

The objectives are clear. This is a modeling exercise without a stated expected outcome. The data are derived from other publications, however it's clear what population is included. I am not a modeller, so can't judge directly the methods. There are no ethical concerns.

Reviewer #2: (No Response)

**Results**

-Does the analysis presented match the analysis plan?

-Are the results clearly and completely presented?

-Are the figures (Tables, Images) of sufficient quality for clarity?

Reviewer #1: The analysis presented matches the plan. The results are presented with good clarity-- a few suggestions are included in this review. One aspect of the analysis is omitted from discussion, which is the case in which prevalence of Dengue at or below 9 years of age is greater than 50%. It's probably wise to limit the options presented in detail for discussion, but would seem useful to included some mention of it.

Figures and Tables are clear.

Reviewer #2: (No Response)

**Conclusions**

-Are the conclusions supported by the data presented?

-Are the limitations of analysis clearly described?

-Do the authors discuss how these data can be helpful to advance our understanding of the topic under study?

-Is public health relevance addressed?

Reviewer #1: The conclusions are supported by the analysis. Limitations of the analysis are described clearly, though one should be called out in the introduction and methods more clearly: the use of government costs only as the 'cost' component definitely skews the cost-benefit analysis in an unfavorable way.

Public health relevance is addressed, particularly the very important point that the specificity of tests applied prior to vaccination greatly alters the risks and benefits of the vaccination strategy.

Reviewer #2: (No Response)

**Editorial and Data Presentation Modifications?**

Reviewer #1: The authors state that the code will be available on GITHUB. This is excellent, but from the statement itself, it’s not clear that this satisfies the requirement that the data be made available. It appears that all data are from public sources and most or all are given within the manuscript as well, in which case this could simply be stated.

Line 47, spell-check screening

Line 102, it would seem that evidence for the other side of this coin should also be cited here to complete the argument—that is, the effect of vaccination after prior infection. This seems to be in Table 1.

Line 236, “In the baseline scenario of costs”…. Please spell this baseline out again, so the reader doesn’t have to go back through the manuscript to find it and differentiate from what follows.

Line 244 -- In terms OF?

Line 315 – The point about varying effects depending upon the dominance, year to year, of different strains, is a good one. It would be enlightening to discuss this a bit further—is there a worst-case scenario and best-case scenario that could be described, at least in general terms? It would seem that the worst-case scenario is worth discussing.

General – PE9 = 60% was simulated but is not discussed at all, presumably for simplicity. A statement that ICER would be even more favorable at 60% and a pointer to the figures/tables that are relevant would seem to be appropriate.

General – The limitation of the costs to what the government pays, rather than total societal costs, while reasonable, should be very clearly stated in the introduction and methods, as it significantly decreases the figures used for economic benefit and skews the results of the paper toward an unfavorable decision about vaccination.

Ref 10. Is the author correct?

Reviewer #2: (No Response)

**Summary and General Comments**

Reviewer #1: (No Response)

Reviewer #2: This study was to examine the cost-effectiveness of the strategy of screening + DENV using an agent-based modelling approach in Puerto Rico. It is a well-written manuscript addressing an important question. Most assumptions of the modelling are valid. However, I have following comments for authors to consider. 

1. Line 74: Please describe what is he status quo when discussing agenting-based model. It is not clear whether the status quo was the scenario without vaccine or the scenario with vaccination but without screening.

2. Line 90: Mean prevalence was set at 50%. However, lower and upper bound was set as 0.3 and 0.6, respectively. Please clarify why authors assume a potential asymmetrical distribution of the prevalence. 

3. Line 143 mentioned that the study was conducted from a public health perspective. If I understood correctly, the authors meant to say “health system’s perspective” or “health providers perspective”. According to recommendations for conduct cost-effectiveness analyses in health (https://jamanetwork.com/journals/jama/article-abstract/2552214), it is often recommended to conduct CEA from both societal and health system’s perspectives. Authors needs to further explain this perspective on what has been excluded or included in this perspective. It is not until line 177 did I realize the premature death was accounted for in this study. 

4. Line 153: The cost of vaccine varies substantially; it could be as low as 32.6 to as high as 683.8. The cost of vaccine would be a major determinations of the cost-effectiveness. It would be helpful to explain why there was such a huge variation in Puerto Rico in the discussion section. Will the group purchasing would affects the procurement price of vaccine? 

a. It is not clear how $382 was derived? 

5. Study shows that dengue seroprevalence varies among municipalities (https://journals.plos.org/plosntds/article?id=10.1371/journal.pntd.0002159). Given that the prevalence and sensitivity and specific of screen would affect the cot-effectiveness of the vaccine, it would be helpful to discuss the implication of the strategy of implementation DENV in Puerto Rico in municipalities with different seroprevalence.

PLOS authors have the option to publish the peer review history of their article (what does this mean?). If published, this will include your full peer review and any attached files.

Reviewer #1: No

Reviewer #2: No

Figure Files:

Data Requirements:

Reproducibility:

References

---

## [Editor Report · Decision Letter 1]

20 Apr 2021

Dear Mr. España,

Thank you very much for submitting your manuscript "Cost-effectiveness of Dengue Vaccination in Puerto Rico" for consideration at PLOS Neglected Tropical Diseases. As with all papers reviewed by the journal, your manuscript was reviewed by members of the editorial board and by several independent reviewers. The reviewers appreciated the attention to an important topic. Based on the reviews, we are likely to accept this manuscript for publication, providing that you modify the manuscript according to the review recommendations. 

As Associate Editor, I commend the authors for their careful response to the previous round of reviewer comments, their sophisticated simulations, factually based analyses, range of sensitivity analyses, independent funding and generally well-written manuscript.

Nevertheless, there is one item in the current manuscript in that threatens to mislead readers and policy makers. That concerns the baseline values of test performance. The authors rightly cite the WHO policy that appropriate use of the current dengue vaccine depends on testing to separate potential vaccines between those without prior infection (who risk slight harm from vaccination) from those with prior infection (who would gain substantial protection from vaccination). The authors’ sensitivity analyses note the importance of the accuracy of the test to make this distinction. 

My concern is that the current manuscript gives the misleading impression that an existing or licensed rapid test can make this distinction with the degree of accuracy that the paper hypothesizes in its base case. A 2018 paper in Lancet Global Health noted how tests then available, which were developed for testing persons with suspected dengue following a RECENT illness, would NOT be appropriate to make the necessary distinction for vaccination (Ariën KK, Wilder-Smith A. Dengue vaccine: reliably determining previous exposure. Lancet Global Health. 2018;6(8):e830–1). 

That 2018 paper noted two problems. The existing rapid tests were not sufficiently sensitive for remote infections (thereby decreasing sensitivity) and they could cross-react with Zika (thereby decreasing specificity at the same time). The current manuscript takes its central values from the by Luo et al (Clinical Microbiology and Infection 25 (2019) 659e666). Luo et al repeatedly noted that the literature on the performance of dengue testing was based on ill patients suspected of having dengue, as illustrated by the following excerpts from pp. 661 and 665: 

“No studies examined the performance of dengue RDTs to detect remote previous infection…. Overall, there were no studies that directly evaluated the use of RDTs for determination of dengue serostatus, as all studies examined RDT performance in the context of either all samples from patients with possible DENV infection, and/or a subset from samples of secondary infection or convalescent time-points after recent DENV infection. Although all studies included samples from dengue-endemic areas, none of them provided information on vaccination or infection status of patients for other ﬂaviviruses, all of which may lead to cross-reactivity with dengue serological testing…. Development of new dengue RDTs or modiﬁcation of currently available RDTs may be the most beneﬁcial for vaccination screening.”

The paper presents no evidence that there is a test licensed for use in Puerto Rico addresses these shortcomings. If there is such evidence, I believe the authors should include it in their manuscript. If such evidence does not exist, then I suggest the authors consider acknowledging the constraint and adjusting their manuscript accordingly. They may wish to consider one or more of the following adjustments: First, they could consider their base values as aspirational, so their base values would apply if an when a “Rapid Vaccination Dengue Test” were licensed and available. Second, they could offer and justify more modest values of sensitivity and specificity if existing tests were repurposed for this pre-vaccination testing despite their limitations. Third, they could consider laboratory-based tests instead of rapid tests. These would be more accurate but entail added costs, logistical challenges, and likely lower vaccination rates.

A few minor editorial suggestions follow, each identified by its line number.

19 It would be helpful to express the iatrogenic additional dengue hospitalizations on the same scale (per 1000 population) as the hospitalizations averted.

52 A stronger statement would be that other candidates are showing promising results.

60 a vaccine program

94 in other studies. Note: age 9 was also considered in reference 13

110 “many times” – four times would be more precise

121 consider rewording the sentence to clarify that “compared to those not vaccinated” applies to the entire sentence.

142 To facilitate comparison, it would also be helpful to report the additional hospitalizations per 1,000 population.

153 See discussion above on sensitivity and specificity and choice of a rapid test.

154 we (lower case) assumed

329 [Zeng] omitted the cost of screening. Suggest changing to: “simulated universal screening without screening.”

358 it would be unfeasible [suggest changing to “infeasible.”]

446 Consider updating the citation as this paper is now published.

468 cost-benefit thresholds

Sincerely,

Donald S Shepard, PhD, FASTM

Associate Editor

Sincerely,

Donald S Shepard

Associate Editor

Sergio Recuenco

Deputy Editor

As Associate Editor, I commend the authors for their careful response to the previous round of reviewer comments, their sophisticated simulations, factually based analyses, range of sensitivity analyses, independent funding and generally well-written manuscript.

Nevertheless, there is one item in the current manuscript in that threatens to mislead readers and policy makers. That concerns the baseline values of test performance. The authors rightly cite the WHO policy that appropriate use of the current dengue vaccine depends on testing to separate potential vaccines between those without prior infection (who risk slight harm from vaccination) from those with prior infection (who would gain substantial protection from vaccination). The authors’ sensitivity analyses note the importance of the accuracy of the test to make this distinction. 

My concern is that the current manuscript gives the misleading impression that an existing or licensed rapid test can make this distinction with the degree of accuracy that the paper hypothesizes in its base case. A 2018 paper in Lancet Global Health noted how tests then available, which were developed for testing persons with suspected dengue following a RECENT illness, would NOT be appropriate to make the necessary distinction for vaccination (Ariën KK, Wilder-Smith A. Dengue vaccine: reliably determining previous exposure. Lancet Global Health. 2018;6(8):e830–1). 

That 2018 paper noted two problems. The existing rapid tests were not sufficiently sensitive for remote infections (thereby decreasing sensitivity) and they could cross-react with Zika (thereby decreasing specificity at the same time). The current manuscript takes its central values from the by Luo et al (Clinical Microbiology and Infection 25 (2019) 659e666). Luo et al repeatedly noted that the literature on the performance of dengue testing was based on ill patients suspected of having dengue, as illustrated by the following excerpts from pp. 661 and 665: 

“No studies examined the performance of dengue RDTs to detect remote previous infection…. Overall, there were no studies that directly evaluated the use of RDTs for determination of dengue serostatus, as all studies examined RDT performance in the context of either all samples from patients with possible DENV infection, and/or a subset from samples of secondary infection or convalescent time-points after recent DENV infection. Although all studies included samples from dengue-endemic areas, none of them provided information on vaccination or infection status of patients for other ﬂaviviruses, all of which may lead to cross-reactivity with dengue serological testing…. Development of new dengue RDTs or modiﬁcation of currently available RDTs may be the most beneﬁcial for vaccination screening.”

The paper presents no evidence that there is a test licensed for use in Puerto Rico addresses these shortcomings. If there is such evidence, I believe the authors should include it in their manuscript. If such evidence does not exist, then I suggest the authors consider acknowledging the constraint and adjusting their manuscript accordingly. They may wish to consider one or more of the following adjustments: First, they could consider their base values as aspirational, so their base values would apply if an when a “Rapid Vaccination Dengue Test” were licensed and available. Second, they could offer and justify more modest values of sensitivity and specificity if existing tests were repurposed for this pre-vaccination testing despite their limitations. Third, they could consider laboratory-based tests instead of rapid tests. These would be more accurate but entail added costs, logistical challenges, and likely lower vaccination rates.

A few minor editorial suggestions follow, each identified by its line number.

19 It would be helpful to express the iatrogenic additional dengue hospitalizations on the same scale (per 1000 population) as the hospitalizations averted.

52 A stronger statement would be that other candidates are showing promising results.

60 a vaccine program

94 in other studies. Note: age 9 was also considered in reference 13

110 “many times” – four times would be more precise

121 consider rewording the sentence to clarify that “compared to those not vaccinated” applies to the entire sentence.

142 To facilitate comparison, it would also be helpful to report the additional hospitalizations per 1,000 population.

153 See discussion above on sensitivity and specificity and choice of a rapid test.

154 we (lower case) assumed

329 [Zeng] omitted the cost of screening. Suggest changing to: “simulated universal screening without screening.”

358 it would be unfeasible [suggest changing to “infeasible.”]

446 Consider updating the citation as this paper is now published.

468 cost-benefit thresholds

Sincerely,

Donald S Shepard, PhD, FASTM

Associate Editor

Figure Files:

Data Requirements:

Reproducibility:

References

---

## [Editor Report · Decision Letter 2]

29 Jun 2021

Dear Mr. España,

We are pleased to inform you that your manuscript 'Cost-effectiveness of Dengue Vaccination in Puerto Rico' has been provisionally accepted for publication in PLOS Neglected Tropical Diseases.

Best regards,

Donald S Shepard

Associate Editor

Sergio Recuenco

Deputy Editor

1 Regarding ref 25 (Medina and Munoz), I suggest that the authors expand the citation to include the reference to the presentation at ACIP and the web link for downloading the presentation.

2 Regarding all references, the authors may wish to check the PLOS style and reformat as needed (e.g. order of first and last names, sentence capitalization of the title of the article and title capitalization of the name of journal).

---

## [Editor Report · Acceptance letter]

19 Jul 2021

Dear Mr. España,

We are delighted to inform you that your manuscript, "Cost-effectiveness of Dengue Vaccination in Puerto Rico," has been formally accepted for publication in PLOS Neglected Tropical Diseases.

Best regards,

Shaden Kamhawi

co-Editor-in-Chief

Paul Brindley

co-Editor-in-Chief
